# Association of Low Alanine Aminotransferase Values with Extubation Failure in Adult Critically Ill Patients: A Retrospective Cohort Study

**DOI:** 10.3390/jcm10153282

**Published:** 2021-07-25

**Authors:** Yoav Weber, Danny Epstein, Asaf Miller, Gad Segal, Gidon Berger

**Affiliations:** 1Department of Internal Medicine “B”, Rambam Health Care Campus, Haifa 3109601, Israel; danyep@gmail.com (D.E.); g_berger@rambam.health.gov.il (G.B.); 2Critical Care Division, Rambam Health Care Campus, Haifa 3109601, Israel; 3Medical Intensive Care Unit, Rambam Health Care Campus, Haifa 3109601, Israel; asafmiller@gmail.com; 4Department of Internal Medicine “T”, Chaim Sheba Medical Center, Tel-Hashomer, Ramat Gan 6971039, Israel; Gad.Segal@sheba.health.gov.il; 5Sackler Faculty of Medicine, Tel-Aviv University, Ramat-Aviv 6997801, Israel; 6Ruth and Bruce Rappaport Faculty of Medicine, Technion-Israel Institute of Technology, Haifa 3109601, Israel

**Keywords:** alanine aminotransferase, respiration, artificial, sarcopenia, ventilator weaning, mechanical ventilator weaning, respiratory insufficiency

## Abstract

Background: Liberation from mechanical ventilation is a cardinal landmark during hospitalization of ventilated patients. Decreased muscle mass and sarcopenia are associated with a high risk of extubation failure. A low level of alanine aminotransferase (ALT) is a known biomarker of sarcopenia. This study aimed to determine whether low levels of ALT are associated with increased risk of extubation failure among critically ill patients. Methods: This was a retrospective single-center cohort study of mechanically ventilated patients undergoing their first extubation. The study’s outcome was extubation failure within 48 h and 7 days. Multivariable logistic and Cox regression were performed to determine whether ALT was an independent predictor of these outcomes. Results: The study included 329 patients with a median age of 62.4 years (IQR 48.1–71.2); 210 (63.8%) patients were at high risk for extubation failure. 66 (20.1%) and 83 (25.2%) failed the extubation attempt after 48 h and 7 days, respectively. Low ALT values were more common among patients requiring reintubation (80.3–61.5% vs. 58.6–58.9%, *p* < 0.002). Multivariable logistic regression analysis identified ALT as an independent predictor of extubation failure at 48 h and 7 days. ALT ≤ 21 IU/L had an adjusted hazard ratio (HR) of 2.41 (95% CI 1.31–4.42, *p* < 0.001) for extubation failure at 48 h and ALT ≤ 16 IU/L had adjusted HR of 1.94 (95% CI 1.25–3.02, *p* < 0.001) for failure after 7 days. Conclusions: Low ALT, an established biomarker of sarcopenia and frailty, is an independent risk factor for extubation failure among hospitalized patients. This simple laboratory parameter can be used as an effective adjunct predictor, along with other weaning parameters, and thereby facilitate the identification of high-risk patients.

## 1. Introduction

Acute respiratory failure is the most common indication for intensive care unit (ICU) admission [1]. Invasive mechanical ventilation (MV) is the cornerstone of management of these patients [2]. Extubation is a crucial moment in the clinical course of many critically ill patients. Despite meeting all predefined criteria and a successful spontaneous breathing trial (SBT), failure of planned extubation still occurs in 10–20% of cases. Extubation failure is associated with prolonged hospital and ICU stays and increased morbidity and mortality. Studies show that extubation failure and reintubation can directly worsen patient outcomes independently of their underlying morbidity [3,4].

Efficient and rapid strategies for identifying patients at high risk of extubation failure are essential to improve the management of weaning and extubation [5]. Several interventions have been shown to reduce the need for reintubation in this population [4]. These include SBT optimization, performance of additional measurements such as central venous oxygen saturation prior to MV liberation, and usage of post-extubation NIV and/or HFNC [4,6,7].

Previous studies have shown that decreased muscle mass and sarcopenia are associated with extubation failure and difficulty in MV liberation [8,9,10]. These studies evaluated muscle mass using measurement of different skeletal muscle diameters on abdominal computed tomography (CT). However, CT-based analysis of body muscle mass has some prominent limitations such as radiation exposure, high cost, and the need to transport the patient outside of the ICU. Others have suggested using bioelectrical impedance technology for rapid and bedside identification of patients with low body mass [11]. Although promising, this technology is not readily available in most ICUs. Ultrasonographic measurements and dual X-ray absorptiometry have also been used in some clinical trials [10].

Alanine aminotransferase (ALT), formerly known as serum glutamate-pyruvate transaminase, is a transaminase enzyme found in plasma, muscle tissue, and the liver. It is routinely measured in hospitalized patients as a part of liver panel tests. Generally, high ALT values (>40 IU/L) are considered pathological and reflect liver damage, caused by various mechanisms, such as hepatitis. A low ALT level in the peripheral blood is associated with low muscle mass [12]. Previous studies have demonstrated a correlation between ALT values and muscle mass, as assessed by CT measurements [13]. Recent studies in different clinical fields have shown that low ALT is a reliable biomarker for sarcopenia [12,14,15,16].

We hypothesized that low ALT blood levels, a known biomarker for sarcopenia, might also be associated with an increased risk of extubation failure in critically ill medical mechanically ventilated patients.

## 2. Methods

### 2.1. Study Design and Data Sources

We conducted a retrospective cohort study, using the data of patients hospitalized in an 8-bed medical intensive care unit (MICU) in Rambam Health Care Campus (Rambam) between 21 November 2016 and 10 June 2020. The medical intensive care unit is a branch of the Division of Internal Medicine. Rambam, located in Haifa, Israel, a 1000-bed tertiary academic hospital serving over 2 million residents of the north of the country. According to hospital records, there are 80,000–90,000 inpatient admissions every year.

The study was approved by the Institutional Review Board at RHCC (approval number RMB-499-19). The need for written informed consent was waived due to the retrospective study design.

Data analyzed in this study were retrieved from Prometheus, a proprietary integrated electronic medical records system developed by Rambam. Data were collected from patient files using the MDClone system (Beer-Sheba, Israel). MDClone system extracts data from electronic medical records, including patient hospitalizations, coded diagnoses, medications, surgical and other procedures, laboratory tests, demographics, and administrative information; all data are presented in a standardized format. The system enables the retrieval of a wide range of variables, for a defined time frame, around an index event [17,18,19,20].

### 2.2. Participants

Orally intubated adult patients hospitalized in Rambam’s MICU were eligible for inclusion in the study if their treating senior ICU physician had recorded them as being ready for primary extubation.

Patients were excluded from the study if a primary tracheostomy (without extubation attempt) was performed. Patients for whom a withdrawal of care decision was performed after extubation, and who died during the 7-day period after extubation attempt, were also excluded. In addition, patients with end-stage renal disease (ICD10 code N18.6) may have low ALT values not associated with whole-body muscle mass; they were therefore excluded from the study [21]. Subjects with chronic liver diseases (ICD10 codes K70-K77) or elevated ALT value (>40 IU/L) were also excluded.

In Rambam’s MICU, ALT is routinely measured daily, during the weekdays, for all the patients. As we did not expect that intubated patients would have an increase in muscle while sedated and mechanically ventilated, we included the lowest ALT value during the 10-day preceding an extubation trial.

### 2.3. Outcome Measures and Variables

In the literature, extubation failure is defined as an inability to sustain spontaneous breathing after extubation within a specified period: from 24–72 h and up to seven days [22,23]. Therefore, the outcomes of our study were both short-term (48 h) and long-term (7 days) extubation success, defined as not requiring re-intubation.

All airway management decisions were at the discretion of the treating intensivist. A checklist used for assessment of weaning readiness is presented in Table 1. During the study period, spontaneous breathing trial was performed with pressure support ventilation of 5–7 cm H_2_O above positive end-expiratory pressure for 20–30 min. T-piece spontaneous breathing trial was not utilized. In our study cohort, all extubations were performed or approved by one of two senior intensivists working in the MICU. All the patients were treated by physiotherapist immediately after extubation. High-risk patients were routinely extubated to NIV or HFNC at the discretion of the treating intensivist. We defined high-risk extubation patients as those older than 65 years or suffering from any underlying chronic cardiac or lung disease. Underlying chronic cardiac diseases included a history of ischemic heart disease or heart failure [24]. Underlying chronic lung diseases included chronic obstructive pulmonary disease. Very high-risk patients were defined as those with two or more risk factors [6,24,25].

Information on patient demographics, body mass index (BMI), admission Acute Physiology and Chronic Health Evaluation II (APACHE-II) score, comorbidities, indication for mechanical ventilation, duration of mechanical ventilation prior to extubation trial, laboratory values, fraction of inspired oxygen (FiO_2_), and vital signs, before extubation, and minimal ALT and creatine phosphokinase values over the preceding 10 days were collected. As some studies found an association between exposure to corticosteroids, vasopressors, muscle relaxants, hypoalbuminemia, and hyperglycemia to neuromuscular dysfunction in ICU patients, these data were also collected [26,27,28,29]. Time (hours) from extubation to re-intubation was recorded up to seven days after extubation.

### 2.4. Statistical Analysis

Patient characteristics were summarized with descriptive statistics. Mean (standard deviation, SD) and median (interquartile range, IQR) were used for the description of normally and non-normally distributed quantitative variables, respectively. Distribution normality was determined using histograms. Bivariable analysis was done to assess candidate variables as risk factors for extubation failure, and the associations between potential risk factors and the outcome were quantified by the odds ratio (OR) and 95% confidence interval (CI). The optimum cutoff for ALT was selected by identifying the value that maximized Youden’s J statistic (sum of sensitivity and specificity) on the Receiver Operating Characteristic (ROC) curve analysis [30]. We used a backward stepwise selection procedure for multivariable logistic regression analyses and included any covariates with a *p* value of  <0.10 on univariate analysis. Multicollinearity between ALT and all the other independent variables was assessed using the variance inflation factor (VIF). A VIF greater than five was considered suggestive for multicollinearity. Missing data were handled using list-wise deletion. All the available data were used for graph generation. Cox regression analysis was performed to generate adjusted survival curves according to the ALT categories (the cutoff value was determined by Youden’s J statistic). Hazard ratio (HR) was calculated. *p* < 0.05 was considered statistically significant. All available data from Rambam’s MICU database within the study time frame were used.

Data analysis was conducted with Statistical Package for the Social Sciences, version 23.0 (IBM SPSS Statistics for Windows, Version 23.0. Armonk, NY, USA: IBM Corp) and Microsoft Excel version 14.0 (Microsoft Corporation, Redmond, WA, USA).

## 3. Results

During the study period, 747 orally intubated and mechanically ventilated patients were hospitalized in Rambam’s MICU and evaluated for extubation readiness. Thirty-seven patients did not fulfill the criteria (see Table 1) for extubation and a primary tracheostomy was performed. Additional 381 patients were excluded from the analyses: 109 due to chronic medical conditions that may interfere with ALT measurement (ESRD and chronic hepatitis/cirrhosis); seven patients died during the 7-day period after extubation attempt (withdrawal of care); and 265 were excluded due to ALT measurements that were either not performed or which were above 40 IU/L (Figure 1).

The final analyses included 329 patients. Table 2 presents the demographic, clinical, and laboratory characteristics of the study cohort. The median study cohort age was 62.4 years (IQR 48.1–71.2), of which 56 (17%) were 75 years old or older. Most of the patients were male (*n* = 205; 62.3%). The mean admission APACHE-II score for the cohort was high, 27.8 (SD 6.5) points, 210 patients (63.8%) were at high risk of extubation failure and 144 (43.8%) at very high risk.

### 3.1. Short-Term (48 h) Extubation Failure

During the first 48 h, 66 patients (20.1%) failed an extubation trial and were re-intubated. The ROC curve for ALT’s prediction of short-term extubation failure is presented in Figure 2a. The area under the ROC curve was 0.62 (95% CI 0.56–0.67, *p* = 0.002) and the optimal cutoff point for ALT was 21 IU/L with sensitivity of 80.3% and specificity of 41.4%. ALT ≤ 21 was recorded among 80.3% (53/66) of patients with failed extubation compared to 58.6% (154/263) in the successfully extubated group (*p* = 0.002). Demographic, clinical, and laboratory characteristics in relation to extubation outcome are presented in Table 3. The results of the bivariate analysis of the correlation between patients’ characteristics and 48 h extubation outcome are shown in Appendix A. Multivariable logistic regression analysis identified albumin, potassium, and ALT as the only independent predictors of the need for re-intubation, with adjusted ORs of 0.66 (95% CI 0.43–0.99, *p* = 0.049), 0.51 (95% CI 0.29–0.88, *p* = 0.02), and 0.97 (95% CI 0.94–0.99, for each IU/L increase, *p* = 0.03), respectively. The AUC of the model was 0.66 (95% CI 0.61–0.71, *p* < 0.001). We used the Cox regression model to generate adjusted survival curves for ALT above and below 21 IU/L subgroups, (*p* < 0.001), Figure 3a. The adjusted HR for low ALT group was 2.41 (95% CI 1.31–4.42).

### 3.2. Long-Term (7 Days) Extubation Failure

During the seven-day follow-up, 83 patients (25.2%) failed an extubation trial and were re-intubated. The ROC curve for ALT’s prediction of long-term extubation failure is presented in Figure 2b. The area under the ROC curve was 0.61 (95% CI 0.55–0.66, *p* = 0.002) and the optimal cutoff point for ALT was 16 IU/L with sensitivity of 61.5% and specificity of 58.9%. ALT ≤ 16 was recorded among 51 (61.5%) of patients with failed extubation compared to 101 (41.1%) in the successfully extubated group (*p* = 0.002). Demographic, clinical, and laboratory characteristics in relation to extubation outcome are presented in Table 3. The results of the bivariate analysis of the correlation between patients’ characteristics and 7 days extubation outcome are shown in Appendix A. Multivariable logistic regression analysis identified ischemic heart disease (adjusted OR 2.31, 95% CI 1.13–4.71, *p* = 0.02), BUN before extubation (adjusted OR 1.01, 95% CI 1–1.02, *p* = 0.04), and ALT (adjusted OR 0.96, 95% CI 0.93–0.99, for each IU/L increase, *p* = 0.03) as the only independent predictors of the need for re-intubation. As APACHE-II score was recorded in 239 cases (72.6%), only these cases were included in the multivariate analysis. The AUC of the model was 0.71 (95% CI 0.65–0.77, *p* < 0.001). We used the Cox regression model to generate adjusted survival curves for ALT above and below 16 IU/L subgroups, (*p* < 0.001), Figure 3b. The adjusted HR for low ALT group was 1.94 (95% CI 1.25–3.02).

## 4. Discussion

This retrospective study found that low ALT values were independently associated with a higher risk of first attempt extubation failure among critically ill patients hospitalized in medical ICU.

Sarcopenia is defined as low muscle mass, strength, and function. It can occur because of normal aging or secondary to severe illness, malnutrition, or low activity level [31]. As severe illness and malnutrition are commonly found in ICU patients, the prevalence of sarcopenia in this population is very high [9]. ICU related sarcopenia is usually a result of critical illness neuromyopathy; although it primarily affects the lower limbs, it is associated with respiratory muscle and diaphragm weakness [10]. Studies have reported that up to 100% of ICU patients with multi-organ failure develop muscle mass loss during hospitalization [32]. Other studies have found that elderly patients lost 10% of total body muscle mass during only three days of immobility, while one week of complete bed rest is associated with >10% reduction in postural muscle strength in healthy subjects [10]. Respiratory muscle mass loss also develops very quickly; diaphragm wasting is seen as soon as 24 h after intubation and diaphragm thickness decreases by 3–6% every 24 h of MV [33,34]. Major risk factors for ICU related sarcopenia include primary disease severity, and length of MV and ICU stay. Some studies found that older age, female gender, administration of corticosteroids and neuromuscular blockers, need for red blood cell transfusion, hypoalbuminemia, and hyperglycemia are also possible risk factors [10,26,27,28,29].

Since normal function and mass of the respiratory musculature is required for initiation of an inspiratory effort and the ability to maintain spontaneous breathing, it is not surprising that diaphragmatic and other respiratory muscle atrophy and weakness are associated with difficulty in weaning from MV. The association between sarcopenia and extubation failure was recently evaluated by Woo et al. [9]. In adult Korean patients mechanically ventilated for at least seven days, sarcopenia, as assessed by L3 skeletal muscle index on abdominal CT, was associated with extubation failure (OR of 24.38, 95% CI 1–594.86). Using the same method of sarcopenia assessment, Kou et al. [8] found that sarcopenia was associated with prolonged weaning (more than seven days of weaning after the first SBT) or need for reintubation within 48 h after extubation in critically ill surgical patients in Taiwan.

There is no gold standard for sarcopenia assessment in the ICU. Although muscle mass can be measured by either bioelectrical impedance, dual X-ray absorptiometry, CT, magnetic resonance imaging, and ultrasonography, all these methods are preliminary research tools limited to clinical trials [10].

ALT, a simple and inexpensive enzyme measurement commonly included in routine blood tests, is a surrogate marker for low general body muscle mass and sarcopenia [13,16]. It was recently shown that low ALT levels are associated with reduced muscle strength, adverse outcomes in the general population of hospitalized patients, patients with chronic obstructive pulmonary disease exacerbation, and rehabilitation program participants [12,15,16,35]. To the best of our knowledge, this is the first study to assess the correlation between low ALT and extubation failure.

This study retrospectively included all medically critically ill patients who fulfilled all clinical and laboratory criteria for extubation and successfully passed SBT. In this heterogenic group, low ALT was independently associated with both short and long-term extubation failure. Based on the existing evidence, we believe that this association is explained by the extremely low muscle mass of patients with ALT levels below 16–21 IU/L.

In our cohort, the 48 h failure rate was 20% and seven-day extubation failure rate was 25%; this high rate can be explained by the high-risk population included in the cohort (63.8% had a high risk of failure; 43.8% had a very high risk). The previously reported reintubation rates in these groups are between 20% to 30%, or even higher [25].

Prediction of “extubation failure” is essential, as both delayed and failed extubation have significant consequences on patients’ clinical course, such as prolonged MV and ICU stay, need for tracheostomy, increased cost of treatment, and higher mortality [31]. In this study, we suggest a simple, cheap, and readily available laboratory marker that may help physicians to prepare for challenging extubation and offer better preparation, such as SBT optimization and extubation to noninvasive ventilation or high flow nasal cannula.

Our study has some limitations. First, this was a retrospective study with relatively small sample size. Second, we did not include ventilation indexes (such as plateau pressure, respiratory rate, rapid shallow breathing index) before an extubation attempt. However, all the patients included in the cohort were considered suitable for extubation by one of two senior physicians working in our medical ICU according to a built-in checklist. Third, some factors that may have an impact on the extubation outcome, such as sequential organ failure assessment score, performance status, fragility scores, and ejection fraction were unavailable for the study cohort and therefore were not included in the analysis. Fourth, only patients with available and normal serum ALT levels were included. This test was performed at the discretion of the treating physician, as part of routine chemistry analysis, rather than systematically. The usage of ALT as a surrogate of sarcopenia and its ability to predict extubation outcome is limited to patients with low ALT; however, some sarcopenic patients may have elevated ALT due to hepatic damage which is not uncommon in critically ill patients. Fifth, we did not include the exact reason for re-intubation in the analysis. Sixth, we did not include utilization of NIV and HFNC after extubation in our analysis. Although we routinely extubate high risk patient to NIV/HFNC, these methods are utilized at the discretion of the treating physician. Seventh, in our cohort, some well accepted risk factors for extubation failure (such as comorbidities and age) were not associated with extubation failure. This discrepancy may be explained by some methodological differences (such as the definition of extubation failure) between our study and previous studies [24], as well as some differences in the characteristics of patients included in the studies.

## 5. Conclusions

Low ALT values are significantly associated with extubation failure and should be flagged in ventilated patients prior to extubation. Looking at the ALT values throughout the hospitalization, in addition to the evaluation of other objective clinical criteria, could aid the clinician in making a more comprehensive, personalized, and informed decision regarding mechanical ventilation weaning and planning patient management after extubation. Patients with low ALT values should be managed more closely during the weaning process and utilization of strategies proven to reduce reintubation rates, such as HFNC and NIV, should be considered in this population. Additional studies that look at the effect of integration of ALT in risk stratification models are warranted.

## Figures and Tables

**Figure 1 jcm-10-03282-f001:**
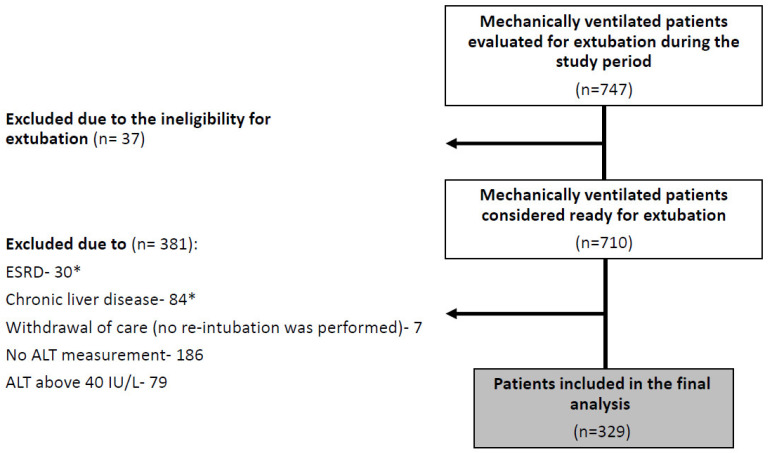
Study flow chart. * 5 patients suffered from both ESRD and chronic liver disease. ESRD—end-stage renal disease; ALT—alanine transaminase.

**Figure 2 jcm-10-03282-f002:**
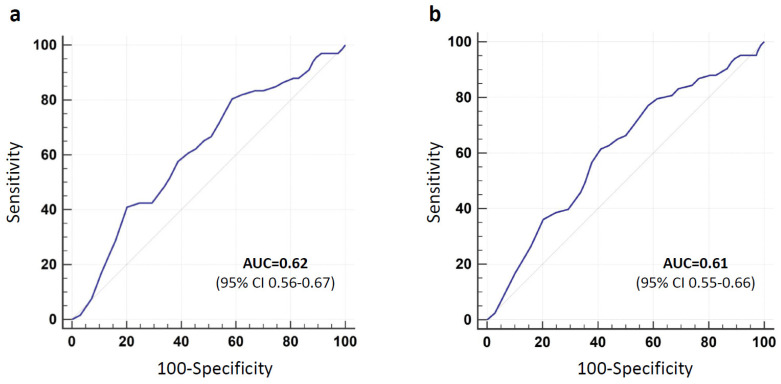
The Receiver Operating Characteristic Curve (ROC) curve for ALT’s prediction of short-term (**a**) and long-term (**b**) extubation failure. AUC—Area under the curve; C—confidence interval.

**Figure 3 jcm-10-03282-f003:**
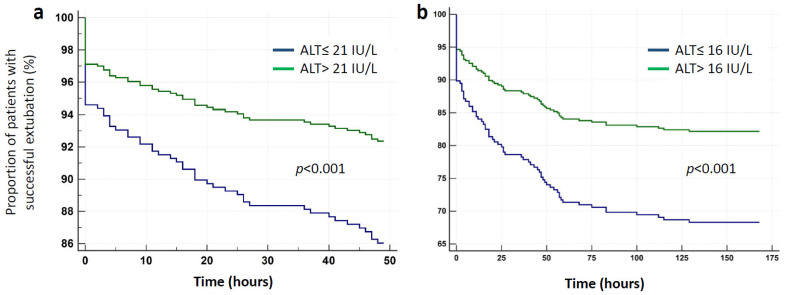
Cox regression survival curves for short-term (**a**) and long-term (**b**) extubation failure according to the lowest ALT value during 10 days preceding extubation trial. Cox regression analysis demonstrated that there were significant differences in extubation failure rates among patients with different ALT values when adjusted for other independent predictors of the need for re-intubation: albumin and potassium for short-term (**a**) and ischemic heart disease and BUN for long-term (**b**) extubation failure. ALT—Alanine transaminase; BUN—Blood urea nitrogen.

**Table 1 jcm-10-03282-t001:** A checklist used for assessment of weaning readiness. These criteria were assessed by a senior physician before extubation (all must be met).

➢ The primary cause of respiratory failure was resolved or significantly improved	
➢ The patient was able to protect the airway, maintain airway patency, had a strong cough, minimal secretions	
➢ Glasgow Coma Scale > 8	
➢ Hemodynamic stability- heart rate 60–130 bpm and systolic blood pressure 90–185 mmHg without or on low-dose vasopressors	
➢ Body temperature between 36 °C and 39 °C	
➢ Respiratory criteria for initiation of spontaneous breathing trial:	
	Positive end-expiratory pressure ≤5 cm H_2_O
	PaO_2_/FiO_2_ > 150
	FiO_2_ < 50%
	pH > 7.25
➢ Criteria of successful spontaneous breathing trials *:	
	Respiratory rate < 35 breaths/minute
	Heart rate < 140 bpm or < 20% change from baseline
	Arterial oxygen saturation > 90% or PaO_2_ > 60 mmHg on FiO_2_ < 50%
	Systolic blood pressure < 180 mmHg or > 80 mmHg or < 20% change from baseline
	No signs of increased work of breathing or distress
	Rapid Shallow Breathing Index < 105 min/L
➢ Positive cuff-leak test	

* Spontaneous breathing trial was performed with pressure support ventilation of 5–7 cm H_2_O above positive end-expiratory pressure for 20–30 min. T-piece spontaneous breathing trial was not utilized during the study period.

**Table 2 jcm-10-03282-t002:** Demographic, clinical, and laboratory characteristics of 329 patients included in the study.

Age, years (IQR)		62.4 (48.1–71.2)
	Elderly (≥75 years), *n* (%)	56 (17%)
Male gender, *n* (%)		205 (62.3%)
Body mass index, kg/m^2^ (IQR)		26.85 (23.5–31)
Comorbidities		
	Diabetes Mellitus, *n* (%)	113 (34.4%)
	Chronic Kidney Disease, *n* (%)	54 (16.4%)
	Congestive Heart Failure, *n* (%)	89 (27.1%)
	Cerebrovascular disease, *n* (%)	52 (15.8%)
	COPD, *n* (%)	75 (22.8%)
	Hypertension, *n* (%)	171 (52%)
	Ischemic Heart Disease, *n* (%)	74 (22.5%)
Indication for mechanical ventilation		
	Neurologic disorders, *n* (%)	88 (26.7%)
	Sepsis, *n* (%)	60 (18.2%)
	Pneumonia, *n* (%)	43 (13.1%)
	Acute exacerbation of COPD, *n* (%)	35 (10.6%)
	Cardiac disorders, *n* (%)	29 (8.8%)
	Shock (non-septic), *n* (%)	12 (3.6%)
	Airway disorder, *n* (%)	8 (2.4%)
	Asthma, *n* (%)	5 (1.5%)
	Other indication, *n* (%)	49 (14.9%)
Admission APACHE-II score (SD) *		27.8 (6.5)
Medications		
	Corticosteroids (during 7 days before extubation), *n* (%)	113 (34.4%)
	Muscle relaxants (during index admission), *n* (%)	60 (18.2%)
	Vasopressors (during 24 h before extubation), *n* (%)	59 (17.9%)
Length of ICU stay before first extubation trial, days (IQR)		3.81 (1.64–7.47)
Duration of mechanical ventilation, days (IQR)		2.86 (1.19–5.87)
High risk of extubation failure, *n* (%)		210 (63.8%)
Very high risk of extubation failure, *n* (%)		144 (43.8%)
Laboratory values (last value before extubation)		
	Albumin, g/dL (IQR)	2.8 (2.48–3.2)
	Creatinine, mg/dL (IQR)	0.78 (0.62–1.27)
	Blood urea nitrogen, mg/dL (IQR)	20 (13–32.6)
	Hemoglobin, g/dL (IQR)	10.2 (8.7–11.9)
	Sodium, mEq/L (IQR)	140 (138–143)
	Potassium, mmol/L (IQR)	3.8 (3.5–4.1)
	Phosphorus, mg/dL (IQR)	3.25 (2.49–4.1)
	Magnesium, mg/dL (IQR)	1.99 (1.76–2.1)
	Brain natriuretic peptide, pg/mL (IQR) **	1296 (396–3711)
	pH	7.41 (7.36–7.47)
	pO_2_, mmHg (IQR)	103 (82–135)
	pCO_2_, mmHg (IQR)	46 (39–53)
	Bicarbonate, mmol/L (IQR)	29.3 (24.9–35.1)
	Lactate, mmol/L (IQR)	1.2 (0.9–1.7)
	Ionized calcium, mmol/L (IQR)	1.1 (1.04–1.14)
	Hyperglycemia (>180 mg/dL) during 48 h preceding extubation trial, *n* (%)	198 (60.2%)
Vital signs (last value before extubation)		
	Heart rate, bpm (IQR)	91 (80–107)
	Systolic blood pressure, mmHg (IQR)	139 (121–157)
	Diastolic blood pressure, mmHg (IQR)	66 (57–75)
	Core Temperature, °C (IQR)	36.8 (36.3–37.4)
	Oxygen saturation, % (IQR)	98 (96–100)
	Fraction of inspired oxygen, % (IQR)	40 (35–40)
Muscle related laboratory values (lowest value over the preceding 10 days)		
	Alanine transaminase, IU/L (IQR)	18 (11–25)
	Creatine phosphokinase, U/L (IQR) ***	132 (51–422)
30-day mortality (after extubation attempt), *n* (%)		28 (8.5%)

* APACHE-II score was recorded in 239 cases (72.6%); ** Brain natriuretic peptide values were available for 72 patients (21.9%); *** Creatine phosphokinase values were available for 163 patients (49.5%); OR—odds ratio; 95% CI—95% confidence interval; IQR—interquartile range; COPD—chronic obstructive lung disease; APACHE-II score—Acute Physiology and Chronic Health Evaluation II; ICU—intensive care unit.

**Table 3 jcm-10-03282-t003:** Demographic, clinical, and laboratory characteristics of 329 patients included in the study, in relation to short-term (48 h) and long-term (7 days) extubation outcomes.

		Short-Term (48 h) Extubation Outcome		Long-Term (7 Days) Extubation Outcome	
		Failure (*n* = 66)	Success (*n* = 263)	Failure (*n* = 83)	Success (*n* = 246)
Age, years (IQR)		63.6 (54.3–75.4)	62.3 (46.3–71)	65.7 (55–76.2)	62.1 (45.7–70)
Male gender, *n* (%)		34 (51.5%)	171 (65%)	45 (54.2%)	160 (65%)
Body mass index, kg/m^2^ (IQR)		27.3 (24.2–29.3)	26.7 (23.4–31.2)	27.5 (24.2–30.9)	26.2 (23.4–31.03)
Comorbidities					
	Diabetes Mellitus, *n* (%)	26 (39.4%)	87 (33.1%)	35 (42.2%)	78 (31.7%)
	Chronic Kidney Disease, *n* (%)	10 (15.2%)	44 (16.7%)	14 (16.9%)	40 (16.3%)
	Congestive Heart Failure, *n* (%)	20 (30.3%)	69 (26.2%)	26 (31.3%)	63 (25.6%)
	Cerebrovascular disease, *n* (%)	12 (18.2%)	40 (15.2%)	15 (18.1%)	37 (15%)
	COPD, *n* (%)	10 (15.2%)	65 (24.7%)	14 (16.9%)	61 (24.1%)
	Hypertension, *n* (%)	34 (51.5%)	137 (52.1%)	48 (57.3%)	123 (50%)
	Ischemic Heart Disease, *n* (%)	16 (24.2%)	58 (22.1%)	26 (31.3%)	48 (19.5%)
Admission APACHE-II score (SD) *		28 (23–33)	28 (23–32)	29.29 (6.66)	27.24 (6.31)
Medications					
	Corticosteroids (during 7 days before extubation), *n* (%)	26 (39.4%)	87 (33.1%)	32 (38.6%)	81 (32.9%)
	Muscle relaxants (during index admission), *n* (%)	12 (18.2%)	48 (18.3%)	17 (20.5%)	43 (17.5%)
	Vasopressors (during 24 h before extubation), *n* (%)	10 (15.2%)	49 (18.6%)	13 (15.7%)	46 (18.7%)
Length of ICU stay before first extubation trial, days (IQR)		4.68 (2.23–7.35)	3.53 (1.5–7.54)	4.87 (2.63–7.72)	3.34 (1.46–6.77)
Duration of mechanical ventilation, days (IQR)		3.7 (1.73–6.35)	2.78 (1.08–5.7)	3.71 (1.74–6.27)	2.63 (1.01–5.66)
High-risk of extubation failure, *n* (%)		42 (63.6%)	168 (63.9%)	58 (69.9%)	152 (61.8%)
Very high-risk of extubation failure, *n* (%)		30 (45.5%)	114 (43.3%)	41 (49.4%)	103 (41.9%)
Laboratory values (last value before extubation)					
	Albumin, g/dL (IQR)	2.65 (2.3–3)	2.9 (2.5–3.2)	2.7 (2.3–3)	2.9 (2.5–3.2)
	Creatinine, mg/dL (IQR)	0.86 (0.6–1.47)	0.77 (0.63–1.24)	0.94 (0.61–1.6)	0.75 (0.62–1.18)
	Blood urea nitrogen, mg/dL (IQR)	25 (14–43.2)	19.7 (12.7–32)	27.9 (16.4–45.3)	19 (12.3–29)
	Hemoglobin, g/dL (IQR)	9.65 (8.5–11.9)	10.3 (8.8–11.9)	9.3 (8.4–11.48)	10.4 (8.9–11.9)
	Sodium, mEq/L (IQR)	140 (137–145)	140 (138–143)	141 (138–145)	140 (138–143)
	Potassium, mmol/L (IQR)	3.7 (3.4–4)	3.8 (3.5–4.2)	3.7 (3.4–4)	3.8 (3.5–4.2)
	Phosphorus, mg/dL (IQR)	3.07 (2.4–3.9)	3.35 (2.52–4.11)	3.14 (2.53–4.02)	3.29 (2.48–4.11)
	Magnesium, mg/dL (IQR)	1.93 (1.71–2.14)	1.99 (1.77–2.24)	1.96 (1.72–2.19)	1.99 (1.76–2.23)
	Brain natriuretic peptide, pg/mL (IQR) **	1868 (301–3347)	1230 (432–3875)	2089 (316–3823)	1182 (425–3711)
	pH	7.41 (7.35–7.46)	7.41 (7.36–7.47)	7.42 (7.34–7.47)	7.41 (7.36–7.47)
	pO_2_, mmHg (IQR)	93 (78–123)	106 (84–138)	96 (80.5–124.5)	106 (83–138)
	pCO_2_, mmHg (IQR)	48 (42–56)	45 (39–53)	47 (41–54)	45 (39–53)
	Bicarbonate, mmol/L (IQR)	30.6 (25.8–36.4)	29.1 (24.8–34.8)	30.6 (24.8–36.4)	29 (24.9–34.8)
	Lactate, mmol/L (IQR)	1.2 (0.95–1.65)	1.1 (0.85–1.7)	1.2 (0.9–1.7)	1.1 (0.8–1.7)
	Ionized calcium, mmol/L (IQR)	1.1 (1.04–1.16)	1.09 (1.04–1.14)	1.1 (1.03–1.16)	1.09 (1.04–1.14)
	Hyperglycemia (>180 mg/dL) during 48 h preceding extubation, *n* (%)	37 (56.1%)	161 (61.2%)	52 (62.7%)	146 (59.4%)
Vital signs (last value before extubation)					
	Heart rate, bpm (IQR)	91 (80–110)	92 (81–107)	91 (80–109)	92 (81–107)
	Systolic blood pressure, mmHg (IQR)	141 (123–156)	139 (121–158)	142 (123–156)	138 (121–158)
	Diastolic blood pressure, mmHg (IQR)	67 (57–75)	66 (57–75)	64 (56–74)	66 (57–75)
	Core Temperature, °C (IQR)	36.7 (36.2–37.4)	36.8 (36.3–37.4)	36.7 (36.2–37.4)	36.9 (36.3–37.4)
	Oxygen saturation, % (IQR)	98 (95–100)	98 (96–100)	98 (96–100)	98 (96–100)
	Fraction of inspired oxygen, % (IQR)	40 (35–40)	40 (35–40)	40 (35–40)	40 (35–40)
Muscle related laboratory values (lowest value over the preceding 10 days)					
	Alanine transaminase, IU/L (IQR)	14 (9–21)	19 (12–26)	15 (9–21)	20 (12–26)
	Creatine phosphokinase, U/L (IQR) ***	105 (38–234)	136 (52–475)	98 (38–257)	142 (56–478)

* APACHE-II score was recorded in 239 cases (72.6%); ** Brain natriuretic peptide values were available for 72 patients (21.9%); *** Creatine phosphokinase values were available for 163 patients (49.5%). IQR—interquartile range; COPD—chronic obstructive lung disease; APACHE-II score—Acute Physiology and Chronic Health Evaluation II.

## Data Availability

The datasets used and/or analysed during the current study are available from the corresponding author on reasonable request.

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
