# Peer review of "Association of Low Alanine Aminotransferase Values with Extubation Failure in Adult Critically Ill Patients: A Retrospective Cohort Study"

_jcm, 2021, doi:10.3390/jcm10153282_

Round 1
Reviewer 1 Report
Review to the JCM- 1267551, Association of low ALT levels with extubation failure
This is a retrospective single-center study measuring the extubation failure within seven days among mechanically ventilated patients and its association to ALT levels. The authors concluded that low ALT is an independent risk factor for failure and may facilitate the identification of high-risk patients.
This is a well-written paper – that suggests a relatively simple, cheap, and available lab marker associated with extubation failure. Knowing this fact may help physicians prepare for risky and challenging extubation and offer better preparation and support in the process. This part was not discussed enough.
Methods:
Why did you choose 7 days for extubation failure? Any reference? Reason for this? can you mention your decision in Methods- also discussing why not reporting the short-term failure as well?
As this is a retrospective study – all the criteria for extubation are part of a protocol? Or just adhere to the local practice? Please elaborate- is there a checklist? Other?
SBT was the same to all patients? Don’t you use t-peice from time to time? All of them were confirmed to be on psv?
What were the parameters to pass the SBT? If none or not documented, please specify that it was upon the clinical judgment of the senior physician solely.
Apache on the day of extubation? or on admission?
Can you specify whether – the reason for ICU admission/intubation, total time in the ICU, the administration of paralytics, days on vasopressors were collected as well?
Prolonged mv was defined over 7 days – give a reference –why 7 days?
When discussing high-risk population for reintubation, please add the reference: Thille et al. ccm 2011
In the method, specify whether specific muscle related lab results were also included as data collected: mg/ca/k/na/cpk/myoglobin etc - these are important confounders.
Results
In the Results, the apache is mentioned but not the timing of the apache- also – in the methods section, you specify three factors for describing high-risk patients and in the results, you conclude that 63% were high risk by the apache – please explain.
Discussion:
Before limitations – please explain what clinicians can do with your information :
Wait with extubation?
Extubate to niv? High flow?
In the limitation, you add two senior intensivists as ruling for extubation –was it mentioned in the methods as criteria? Again this is a retrospective study – please explain – is this the protocol for every extubation?
What about controlling for relevant labs? Hypokalemia? Hypocalcemia or hypercalcemia ets?
Time in the ICU before extubation is important as a confounder, the reason for ICU admission and severity of illness on arrival, paralysis also as mentioned.
Can you mention and explain how the risk factors for extubation failure were not significant in your cohort as predictors for extubation failure?
Conclusion:
Again please offer action items for knowing alt is low before extubation…
Or at least areas for future research
Author Response
Dear Reviewer 1,
Please see the attachment.
sincerely
Weber Yoav, MD
Corresponding author

Reviewer 2 Report
Dear the authors
Thank you for your efforts on the manuscripts.
Including the hypothesis, I found that your study was interesting,
However, I have some major and minor concerns about your study.
Major problems
- It is unclear that how many patients were excluded due to the ineligibility for extubation attempts. Based on your data, we are not sure if all enrolled patients were really eligible for extubation attempts. Therefore, I think that at least you need to represent pre-extubation vital signs and arterial blood gas results for enrolled patients in the study.
- I don’t understand why you excluded seven patients who died after the first extubation attempt. I think they may belong to the extubation failure group.
- Important factors for extubation outcome are vital signs (heart rate, respiratory rate, blood pressure) and arterial blood gas before the extubation attempt. Also, performance status (ECOG), organ function (SOFA score) or fragility (Clinical fragility scale) can also have impacts on the extubation outcomes. Hence, I recommend that you investigate these factors and incorporate them into data analyses, particularly multivariable analysis.
- Cardiac dysfunction is another important factor affecting extubation outcome. So, echocardiographic findings (ejection fraction) or BNP or TnI need to be investigated and analyzed.
- In the results, you did not represent ALT levels as means or medians, but only represented proportions of those with low ALT (<17) levels. And, in logistic regression analysis, you did not use continuous variable for ALT. Hence, I wonder how the result would be when you incorporate continuous ALT variables instead of categorical variable (i.e., a low ALT) into the logistic regression analysis.
- Besides, in the multivariable analysis, you used forward stepwise selection method. However, as you know, backward stepwise selection method is more widely used and trustworthy.
- In the Table 2, you indicated that patients with APACHE-II score was 239.
If so, as you know, the number of patients who were included into the logistic analysis will be 239 not 329.
- And, you also said that missing data were handled using list-wise deletion, but in the results, you didn’t mentioned missing values for variables investigated.
- NIV and HF trial after extubation can also affect the extubation outcome, So, I recommend that you investigate the frequency of NIV or HFNC use and incorporate them into the multivariable analysis.
Minor problems
- In the Method, you said that the study is “population-base”. But, I don’t think it is not population-based study.
- This is a retrospective observation study, so I don’t understand why you presented the sample calculation (n = 116). I think the sample calculation is not necessarily needed. Could you explain why?
- The AUC value for ALT levels was not sufficiently high ( = 0.60) and again, you used categorical variable for ALT, not continuous variable, in the multivariable analysis. Hence, you need to mention this as a limitation.
Thank you
Author Response
Dear Reviewer 2,
Please see the attachment.
sincerely
Weber Yoav, MD
Corresponding author

Round 2
Reviewer 1 Report
the authors improved the manuscript. some minor typos
in the results- under long term .... by mistake you write for fig 2b- short term -needed to be fixed to long term
OR of 0.97 for ALT seems to be not impressive -consider writing for every point in ALT
Author Response
Dear
Reviewer 1
Journal of Clinical Medicine
We are pleased to submit the revised version of the manuscript entitled " Association of low alanine aminotransferase values with extubation failure in adult critically ill patients: A retrospective cohort study" (jcm-1267551).
We appreciate the efforts and extensive and constructive comments of the referees. A point-by-point response follows, and a description of all changes to the manuscript has been submitted (the changes are highlighted in yellow in the manuscript).
Reviewer 1
The authors improved the manuscript. Some minor typos.
- In the results- under long term .... by mistake you write for fig 2b- short term -needed to be fixed to long term.
Thank you for your attention, we fixed it to “long-term”.
- OR of 0.97 for ALT seems to be not impressive -consider writing for every point in ALT.
Thank you for this suggestion. Indeed, OR of 0.97 is not impressive. However, in concordance with the suggestion of the referees in the first review round we included ALT as a continuous variable in the logistic regression. In order to clarify the meaning of OR of 0.97 we now emphasize that this is OR for increase in one IU/L. We hope it is clearer now. However, we also provide the hazard ratio for ALT below 16 (short-term outcome) and 21 IU/L (long-term-outcome).
Thank you for your consideration of this revised manuscript.
Sincerely,
Yoav Weber, MD
Corresponding author
Reviewer 2 Report
Dear authors,
Thank you for your great deal of effort on the revision process.
I think that the revised version has been much improved than before.
Sincerely yours
Author Response
Dear
Reviewer 2
Journal of Clinical Medicine
We are pleased to submit the revised version of the manuscript entitled " Association of low alanine aminotransferase values with extubation failure in adult critically ill patients: A retrospective cohort study" (jcm-1267551).
We appreciate the efforts and extensive and constructive comments of the referees. A point-by-point response follows, and a description of all changes to the manuscript has been submitted (the changes are highlighted in yellow in the manuscript).
Reviewer 2
Thank you for your great deal of effort on the revision process. I think that the revised version has been much improved than before.
I thank you very much for your work.
Thank you for your consideration of this revised manuscript.
Sincerely,
Yoav Weber, MD
Corresponding author